# OrgaSegment: deep-learning based organoid segmentation to quantify CFTR dependent fluid secretion
Juliet W. Lefferts[1,2], Suzanne Kroes[1,2], Matthew B. Smith[1,2,3], Paul J. Niemöller[1,2],
Natascha D. A. Nieuwenhuijze[1,2,3], Heleen N. Sonneveld van Kooten[1,2,3], Cornelis K. van der Ent[1],
Jeffrey M. Beekman ●[1,2,3] ✉ & Sam F. B. van Beuningen ●[1,2,3] ✉

Epithelial ion and fluid transport studies in patient-derived organoids (PDOs) are increasingly being used for preclinical studies, drug development and precision medicine applications. Epithelial fluid transport properties in PDOs can be measured through visual changes in organoid (lumen) size. Such organoid phenotypes have been highly instrumental for the studying of diseases, including cystic fibrosis (CF), which is characterized by genetic mutations of the CF transmembrane conductance regulator (CFTR) ion channel. Here we present OrgaSegment, a MASK-RCNN based deep-learning segmentation model allowing for the segmentation of individual intestinal PDO structures from bright-field images. OrgaSegment recognizes spherical structures in addition to the oddly-shaped organoids that are a hallmark of CF organoids and can be used in organoid swelling assays, including the new drug-induced swelling assay that we show here. OrgaSegment enabled easy quantification of organoid swelling and could discriminate between organoids with different CFTR mutations, as well as measure responses to CFTR modulating drugs. The easy-to-apply label-free segmentation tool can help to study CFTR-based fluid secretion and possibly other epithelial ion transport mechanisms in organoids.

Adult and induced pluripotent stem cell-based organoid cultures have emerged as powerful tools to model human disease and treatment in vitro[1]. Adults stem cell (ASC) organoids are grown from primary human tissue and form multicellular 3D epithelial structures that contain single internal lumens[2]. ASC-organoids can be grown from many tissues and are widely used for the study of epithelial diseases, including genetic diseases such as cystic fibrosis, since they harbor the genetic make-up of the donor[3,4].

Cystic fibrosis (CF) affects an estimated 160,000 people worldwide[5] and is caused by mutations in the *cystic fibrosis transmembrane conductance regulator* (*CFTR*) gene[6,7]. The CFTR protein is a cAMP-regulated ion channel that plays a critical role in epithelial ion and fluid transport[8,9]. To date, over 2100 *CFTR* variants have been identified, of which 719 have been classified as CF-causing[8]. These various mutations differentially affect baseline CFTR function and severity of disease expression, as well as the response to CFTR modulator drugs that bind to the mutant CFTR protein and restore its function[9–12]. CFTR modulators are used as single or combination therapy and include the corrector compounds tezacaftor (VX-661) and elexacaftor (VX-445), and the potentiator compound ivacaftor (VX-770). Approximately 85% of people with CF have mutations known to respond to these modulators[5], but identifying individuals who carry rare or ultrarare *CFTR* variants who can also benefit from CFTR modulators remains an important challenge.

Patient derived intestinal organoids (PDIOs) are increasingly being used and validated for the in vitro prediction of individual CFTR residual function and therapy response[13,14]. CFTR function is essential for fluid secretion into the organoid lumen under steady-state conditions and for agonist-induced fluid secretion via forskolin or other cAMP-inducing reagents[13,15]. CF PDIOs can be distinguished from healthy rectal organoids based on steady-state fluid secretion phenotype either by assessing the relative luminal area as fraction of the whole organoids, defined as steady-state lumen area (SLA) or by rectal organoid morphology analysis (ROMA)[15,16]. Additionally, the forskolin-induced swelling (FIS) assay is a

[1]Department of Pediatric Respiratory Medicine, Wilhelmina Children's Hospital, University Medical Center, Utrecht University, 3584 EA Utrecht, The Netherlands. [2]Regenerative Medicine Utrecht, University Medical Center, Utrecht University, 3584 CT Utrecht, The Netherlands. [3]Centre for Living Technologies, Alliance TU/e, WUR, UU, UMC Utrecht, 3584 CB Utrecht, The Netherlands. ✉e-mail: j.beekman@umcutrecht.nl; s.vanbeuningen@umcutrecht.nl

more sensitive method and quantifies CFTR function within the CF disease domain, facilitating discrimination between individuals with CF based on probability of disease features[13,17–20].

All the above-mentioned assays do not allow for the identification of individual organoid structures, i.e. segmentation, and rely on fluorescent labelling of organoids as the oddly-shaped CF-intestinal organoid structures are not recognized by current bright-field based segmentation models, which favour spherical structures[21,22]. Fluorescent labelling is costly, inconvenient and limits the length of organoid observations due to leakage of the dye over-time. A recent bright-field based approach was developed for FIS[23], but this did not allow for individual organoid segmentation and feature extraction. We therefore set out to develop a new model based on bright-field imaging to achieve automated, individual organoid segmentation, and develop new assay formats to complement current CFTR function measurements in PDIOs in the context of highly effective CFTR modulators.

## Results

### Deep-learning based instance segmentation of CF organoids
To optimize accessibility and standardization of functional CFTR measurements in PDIOs, we developed OrgaSegment: a MASK-RCNN based deep-learning segmentation model, which allows for the segmentation of individual organoid structures from bright-field images (Fig. 1a). For the individual segmentation of a wide range of organoid shapes we first created a dataset (231 images), containing various shapes of organoids, imaged using multiple microscopes. All organoids were annotated manually (ground truth) and reviewed using Labelbox by a team of organoid experts[24]. This resulted in a dataset containing a total of 15,515 individually annotated organoids. The dataset was split into a training, validation and evaluation set and used to train a MASK-RCNN model with a transferred learning basis from the MS COCO dataset (Fig. 1b)[25], as described in detail in the methods section. The resulting model was evaluated using the Average Precision (AP) metric at different Intersection over Union (IoU) thresholds against the evaluation dataset containing 12 (974 organoids) grayscale images with ground truth annotations. The mean AP of the different images at an IoU threshold of 0.5 is $0.76 \pm 0.12$(SD) and decreases upon more stringent IoU thresholds (Fig. 1c). Visual inspection of the predictions on the evaluation dataset, which contained various imaging modalities, showed accurate segmentation with some uncertainties at sites of challenging organoid shapes (Fig. 1d). In order to validate the deep-learning segmentation against the current standard, we compared the total segmentation surface of bright-field images with that of the total fluorescent signal (calcein green) measured during forskolin-induced swelling (Fig. 1e). The deep-learning-based segmentation yielded similar organoid surface areas as the conventional fluorescent analysis. Additionally, results from forskolin-induced swelling measurements analysed with OrgaSegment were also comparable to standard calcein green analysis (Supplementary fig. 1). The developed deep-learning segmentation method was wrapped-up in an easy to use tool named OrgaSegment and made available to the general public (https://github.com/Living-Technologies/OrgaSegment).

### *CFTR*-genotype dependent fluid secretion phenotypes of PDIOs under baseline and CFTR modulator incubation conditions
Next, we studied organoid phenotypes to assess fluid secretion, in the context of baseline CFTR function and response to highly effective CFTR modulators, in PDIOs harbouring various *CFTR* mutations. For conventional FIS measurements using fluorescently labelled organoids, organoids are harvested, disrupted into small fragments (2D sheets), plated into 96-well plates and treated with correctors (VX-445/VX-661). After 24 h, organoids are fluorescently labelled and CFTR function is assayed upon forskolin stimulation and potentiator treatment (VX-770)[26]. During the 24 h period, organoid fragments develop into 3D structures and drugs can impact on CFTR protein production and trafficking (Fig. 2a). We monitored individual organoids during the whole process of plating and CFTR function measurement and observed that PDIOs with various *CFTR*

genotypes all had small luminal phenotypes at the time of corrector treatment (t = 0 h), ~ 1 h after organoid plating (Fig. 2a). Lumens remained small for all vehicle-treated organoids during 24 h incubation, but we observed fluid-filled lumens in PDIOs with rare *CFTR* genotypes (PDIO 01, 02 and 03) after 24-hour VX-445/VX-661 incubation (Fig. 2a). This was not observed for PDIOs with F508del/class I mutations, which did show strong luminal fluid secretion upon additional forskolin treatment. We determined the effect of this modulator-induced swelling by expressing the lumen area relative to the total organoid area, as relative lumen area has previously shown to be able to discriminate between CF-PDIOs and healthy rectal organoids[15] (Fig. 2b). Relative lumen areas in F508del/class I PDIOs were below 5% after VX-445/VX-661 incubation and increased to ~60% upon VX-770 and forskolin addition. In contrast, PDIO 01 had a relative lumen area of ~45%, and PDIO 02 and PDIO 03 even ~70–75% after VX-445/VX-661 incubation, indicative of CFTR function within the WT range[15]. Further increase in lumen area upon VX-770 and forskolin addition was not significant for PDIO 02 and PDIO 03 as relative lumen areas only reached ~75%. Relative lumen areas for vehicle-treated donors remained below 10% prior to forskolin addition which confirms residual CFTR function within the CF-range and that the overnight swelling is indeed modulator-induced[15]. This forskolin-independent drug-induced lumen expansion shows that modulator response already occurs during overnight VX-445/VX-661 incubation in some circumstances. However, as only swelling after 24 h is recapitulated in the FIS response[13], this corrector-induced pre-swelling can lead to underestimation of CFTR function restoration when measured by FIS. These data demonstrate that luminal fluid secretion phenotypes of PDIOs can be strongly affected by incubation with VX-445 and VX-661 in a *CFTR* genotype-dependent manner.

### Quantification of drug-induced organoid swelling
We implemented OrgaSegment in a new and automated assay format, based on previous observations that organoids swell upon addition of CFTR modulators[15]. Since we have shown in the previous section that organoids already strongly responded to CFTR modulators within 24 h of treatment, we explored whether OrgaSegment could quantify individual organoid swelling during this incubation period. PDIOs were plated and incubated with CFTR modulators, and bright-field images were acquired and analyzed using OrgaSegment at indicated time points (Fig. 3a). Individual organoid swelling was measured and mean swelling per condition per individual experiment was calculated and used as biological replicate (Fig. 3b). To assess heterogeneity in swelling response, we quantified swelling per individual organoid structure for F508del/class I and F508del/F508del PDIOs in the presence of DMSO or VX-445/VX-661/VX-770 modulator therapy. Swelling distribution shows a heterogeneity in response to modulator treatment (Fig. 3c), as has been observed previously for FIS response, partially due to small or nonviable structures[15]. Next we determined the minimum amount of organoid structures required per experimental condition. We calculated mean swelling of groups of randomly selected organoids, with bin sizes ranging from 2 to 256 structures, and repeated this 10 times. Results show that, on average, a minimum of 32 structures is required for mean swelling and standard deviation to stabilize (Fig. 3d). These results demonstrated that OrgaSegment can measure individual organoid swelling for the quantification of drug-induced swelling (DIS).

### DIS identifies VX-445/VX-661/VX-770 responsive *CFTR* genotypes
Next, we used OrgaSegment to assess functional CFTR restoration in PDIOs with various genotypes. DIS response upon CFTR modulator incubation was measured in multiple donors per genotype (Fig. 4a). For DIS response per donor line see Supplementary fig. 2a. Class I/class I PDIOs did not show DIS response in any of the conditions, as expected[27]. DIS response in F508del/class I and F508del/F508del PDIOs treated with VX-445/VX-661/VX-770 was significant, as is in accordance with clinical response[28,29]. VX-661/VX-770 responses were not significant in these genotypes[30]. DIS response to all modulator combinations was significant in F508del/S1251N

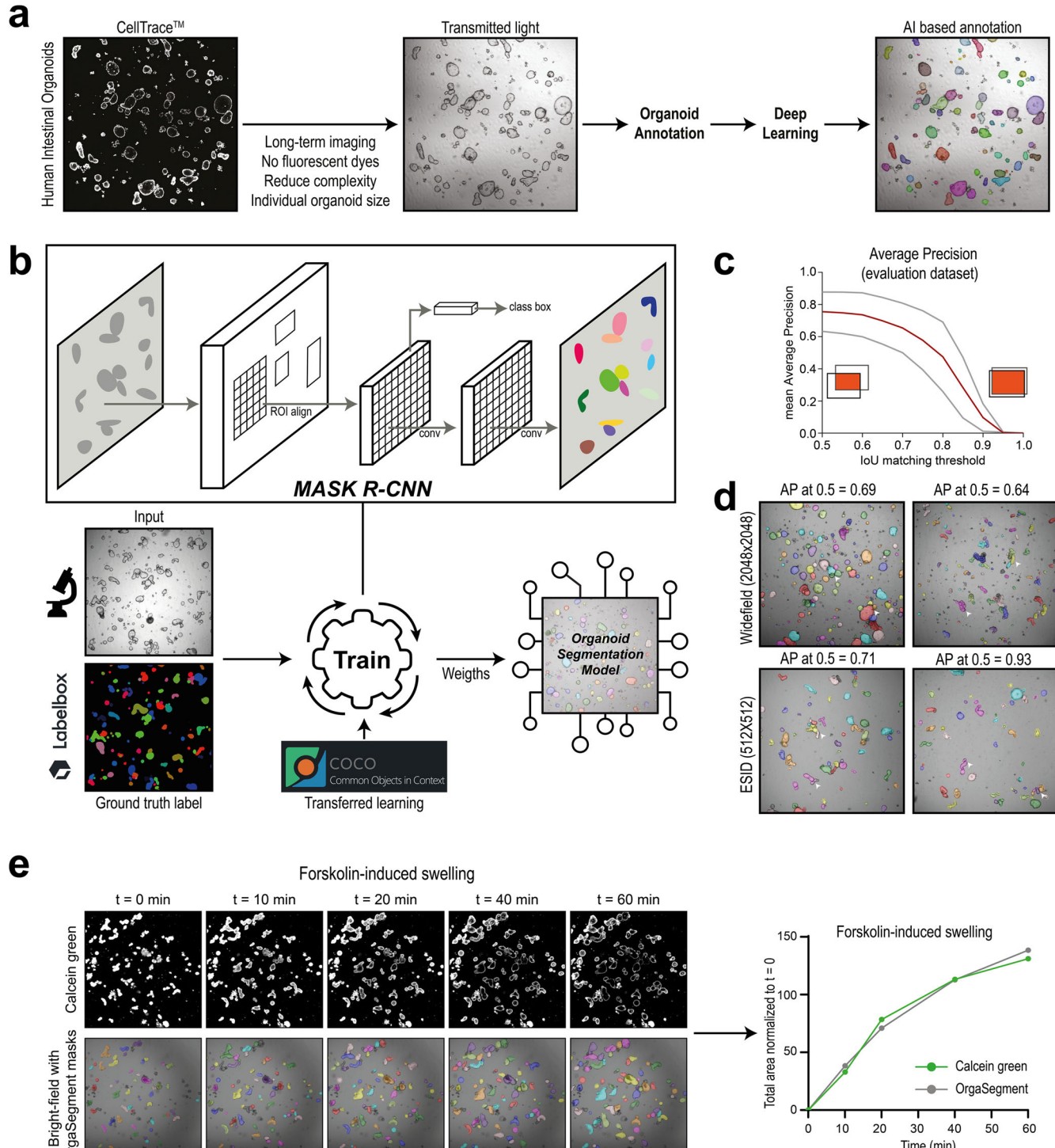

**Fig. 1 | Overview of development and evaluation of the OrgaSegment model.**
**a** Overview of the OrgaSegment rationale and development. **b** Schematic overview of the deep-learning training workflow with the Mask R-CNN framework (adapted from He K et al.[34]). **c** Mean Average Precision (red) with SD (grey) plot of the evaluation dataset at different IoU matching thresholds. **d** Representative images of OrgaSegment based organoid segmentation on the evaluation dataset showing different average precisions. Imaging was performed using transmitted widefield light and 2048 × 2048 pixels (Widefield) or using transmitted laser light and 512 × 512 pixels (ESID). The white arrowheads indicate regions of interest, namely instance segmentation of neighbouring organoids or impartially segmented organoids. **e** Representative intestinal organoid images of calcein green signal and bright-field images with OrgaSegment masks during forskolin-induced swelling. Here, different colour masks do not represent organoid ID and tracking was not applied. Quantification of the surface areas shows the normalized size of the total organoid area from calcein green and OrgaSegment signal.

and F508del/R117H PDIOs[28,29]. Additional overnight 0.128 μM forskolin treatment significantly increased DIS response in F508del/S1251N PDIOs, showing the potential to increase the sensitivity of the DIS assay for lower CFTR function levels (Supplementary fig. 2b). Usage of 0.128 μM forskolin

has previously been shown in the FIS assay to correlate with in vivo CFTR function parameters[19,31]. Overnight swelling with 0.128 μM forskolin, in the absence of CFTR modulators, can also be used to measure residual CFTR function (Supplementary fig. 2b).

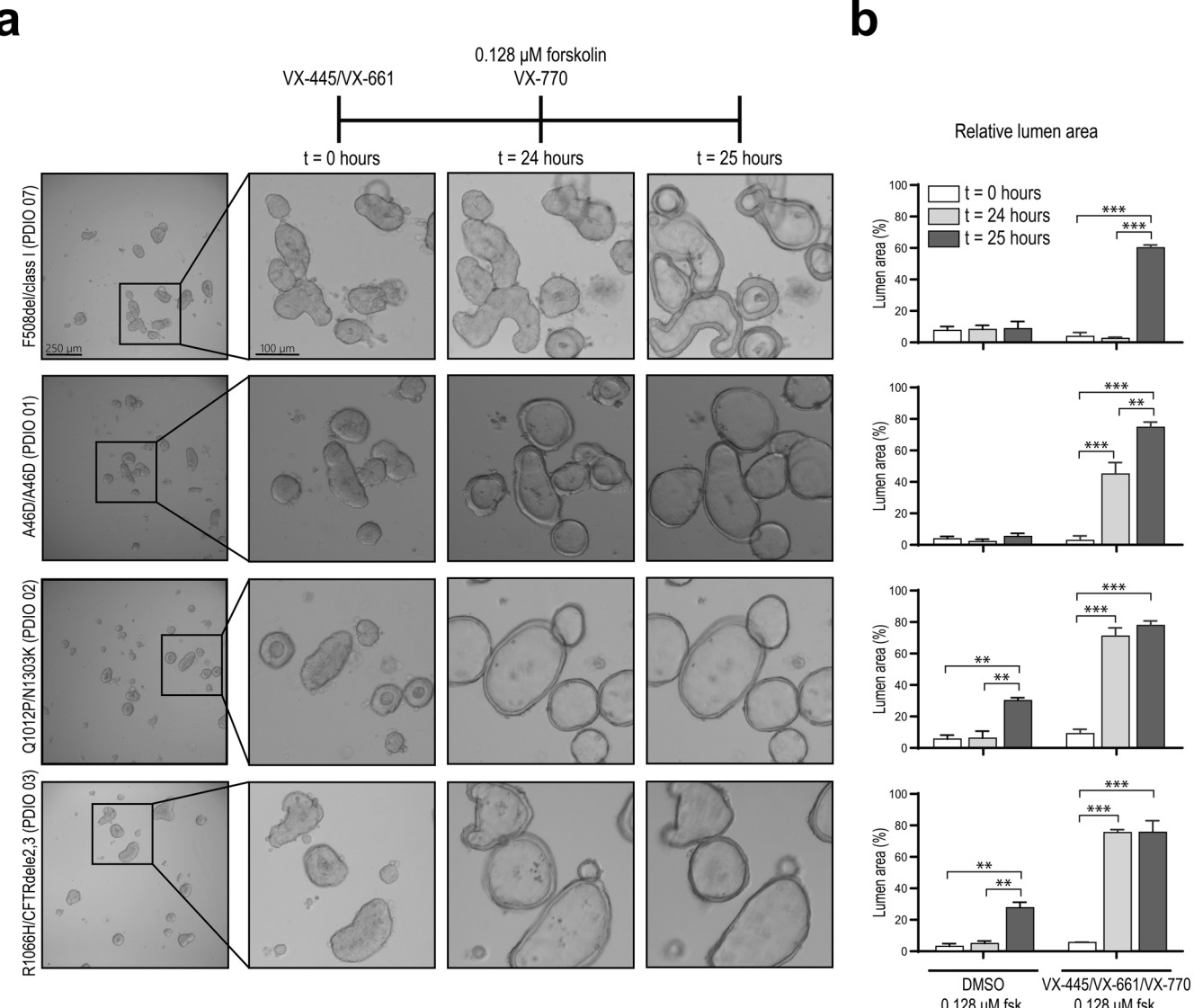

**Fig. 2 | Swelling in response to CFTR modulator treatment in intestinal organoids with various CFTR genotypes. a** Overview of the FIS assay timeline and representative images of PDIOs at various stages of the FIS assay; directly after VX-445/VX-661 addition (t = 0 h), upon VX-770 and forskolin treatment (t = 24 h) and 1 h after forskolin treatment (t = 25 h), for PDIOs with various genotypes. **b.** Quantification of SLA at the various stages of the FIS assay, in the presence of DMSO (control) and VX-445/VX-661/VX-770. Error bars represent standard deviation and significance, calculated using two-tailed t-tests, is indicated where significant, (*$p < 0.05$, **$p < 0.01$, ***$p < 0.001$).

Finally, we measured both FIS (in the presence of 0.128 μM forskolin) and DIS in PDIOs that exhibit luminal fluid secretion upon overnight VX-445/VX-661 incubation (Fig. 4b). FIS and DIS responses were significant in PDIO 01, which displayed moderate luminal swelling, upon VX-445/VX-661 incubation. However, VX-445/VX-661/VX-770 FIS responses in PDIO 02 and PDIO 03 were only ~50–65% of the F508del/class I response. In contrast, DIS levels in PDIO 02 and PDIO 03 showed significant CFTR restoration, comparable to the F508del/class I response to VX-445/VX-661/VX-770 treatment, and in agreement with high levels of luminal fluid secretion prior to addition of forskolin (Fig. 2). Together these results demonstrate that OrgaSegment can be used to measure highly effective CFTR modulation by quantification of individual organoid swelling after 24 h modulator incubation.

## Discussion
Here we present a comprehensive annotated dataset of intestinal organoids together with OrgaSegment, a MASCK-RCNN model for the individual

segmentation of intestinal organoids from bright-field images, and its application for the measurement of functional CFTR restoration in individual PDIOs under various assay conditions.

Models for the segmentation of cells or organoids have been developed before, including OrgaQuant and Cellpose[21,22]. However, OrgaQuant and Cellpose recognize round organoids or favor round objects of similar size, respectively, and are thus unsuitable for the segmentation of CF-PDIOs. In contrast, OrgaSegment allows for the recognition and quantification of individual organoid structures regardless of organoid shape, including the oddly shaped, unswollen CF intestinal organoids. Additionally, OrgaSegment is able to segment organoids that are clustered together as individual structures.

A limitation of the OrgaSegment model is the possibility of biases that could have been introduced, like the microscope type, bright-field imaging modalities, and the organoid type and cultures. For the creation of OrgaSegment we only trained the model on intestinal organoids. However, the model allows for easy adaption by (transferred learning) training of

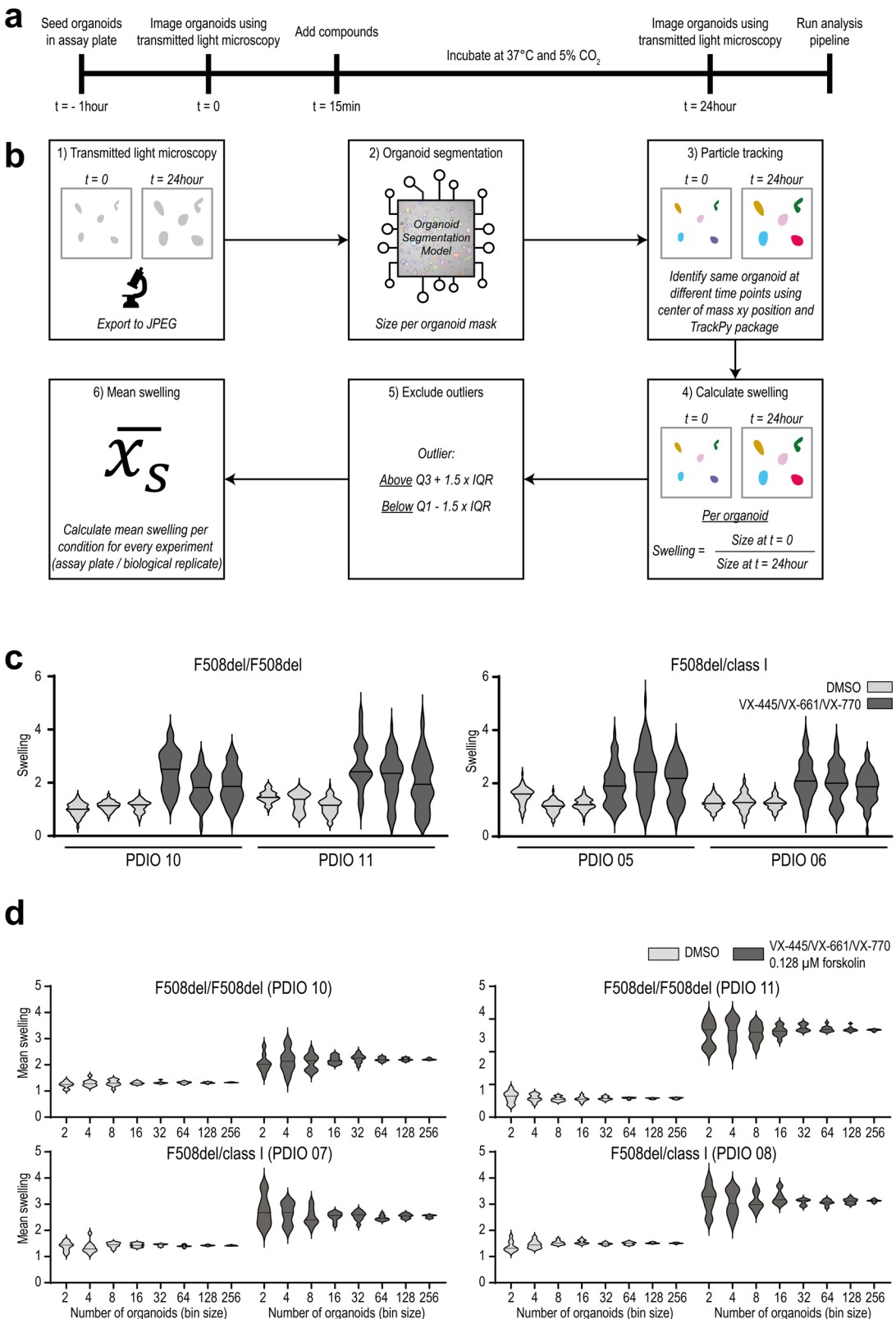

**Fig. 3 | The DIS experimental pipeline and typical swelling distribution of individual organoid structures within an experiment. a** Overview and timeline of a typical DIS experiment. **b** Step-by-step schematic overview of a DIS experiment and analysis. Starting with (1) imaging and image exporting to JPEG, followed by (2) organoid segmentation using OrgaSegment, (3) particle tracking, (4) calculation of swelling per organoid, (5) exclusion of outliers, and finally (6) calculating mean swelling per condition and per experiment (assay plate/biological replicate).

**c** Distribution of organoid swelling for individual F508del/class I and F508del/F508del PDIO structures in the presence of DMSO or VX-445/VX-661/VX-770 treatment. Two different donors are measured per genotype in biological triplicate. **d** Overview of the swelling distribution of randomly selected organoids with various bin sizes, for F508del/class I and F508del/F508del PDIOs in the presence of DMSO or VX-445/VX-661/VX-770 treatment.

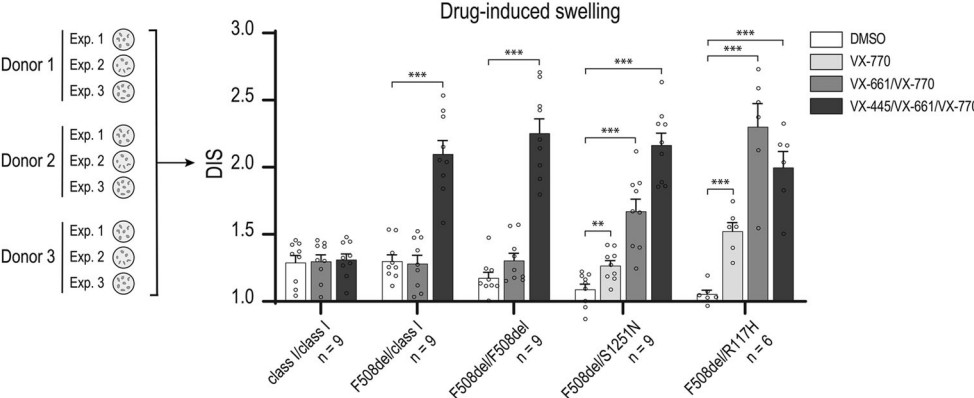

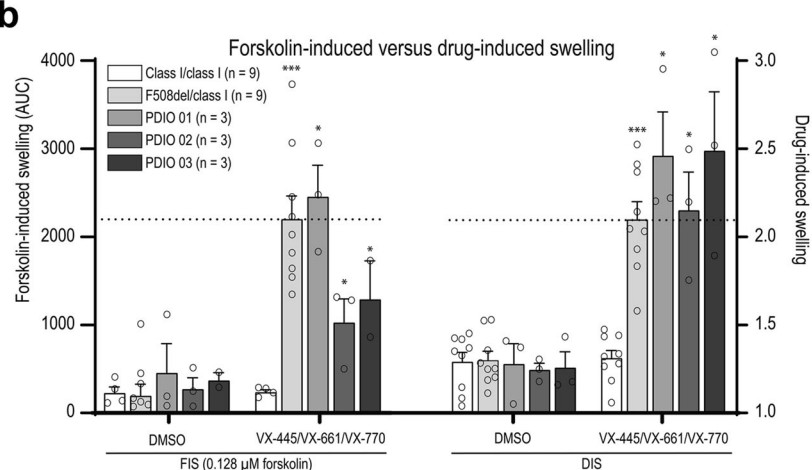

**Fig. 4 | Swelling response upon CFTR modulator treatment in intestinal organoids with various CFTR genotypes, measured with DIS and FIS. a** DIS response to CFTR modulator treatment in PDIOs with various *CFTR* genotypes. Experimental size n equals the amount of donors times three experimental replicates. Error bars represent the standard error of the mean and significance, calculated using two-tailed t-tests, is compared to vehicle (DMSO) treated conditions ($*p < 0.05$, $**p < 0.01$, $***p < 0.001$). **b** FIS and DIS responses of class I/class I and F508del/ class I reference PDIOs (mean of three donors shown) and PDIOs that show preswelling during FIS measurements. FIS is measured at 0.128 μM forskolin. Experimental size n equals the amount of donors times three experimental replicates. Error bars represent the standard error of the mean and significance, calculated using two-tailed t-tests, is compared to vehicle (DMSO) treated conditions ($*p < 0.05$, $**p < 0.01$, $***p < 0.001$).

user-specific datasets and organoid types. Additionally, manual labelling and reviewing of the organoid structures could have introduced a bias in the model. Organoid labelling has been done by a group of over 15 organoid experts to avoid this as much as possible, additionally, quality control of labels has been performed on all images by a reviewer. Furthermore, the manual annotated ground-truth is inherently limited to what is observed by the human eye. We therefore encourage to first test the model on lab specific data before using. The adopted MASK-RCNN framework allows for easy improvement of the OrgaSegment model with organoids from other tissue sources and images from different microscopes, as well as easy adaption for the classification of different organoid types and fluid-secretion models. This makes it a versatile solution for individual organoid segmentation and classification. Implementation of OrgaSegment into CFTR function assays accommodates, in contrast to calcein green usage, label-free and low complex bright-field imaging. Additionally, OrgaSegment allows for automated fine-grade individual organoid measurements, facilitating the correction for individual outliers and technical variation, e.g., the number of organoids per well. This improves assay flexibility, imaging duration and allows for individual organoid analysis, in contrast to previously reported methods, including label-free FIS, ROMA and SLA[15,16,23]. Here we use OrgaSegment to quantify DIS, however OrgaSegment can also be used to quantify PDIOs and potentially other organoid models in other CF and non-CF fluid secretion measurements such as FIS[13].

New, highly effective CTFR modulators lead to challenges for the in vitro quantification of functional CFTR restoration using FIS. We found PDIOs harbouring *CFTR* mutations that are currently ineligible for modulator treatment showing significant luminal fluid secretion upon overnight VX-445/VX-661 incubation, indicative of functional CFTR rescue. This suggests that these rare CFTR variants have a combined trafficking and potentiating defect, that upon restoration by VX-445/VX-661 can be activated in organoids under baseline culture conditions (presumably by low endogenous cAMP). As such the restored CFTR variants functionally acquire activation characteristics associated with wtCFTR as wtCFTR-containing intestinal organoids always have fluid-filled lumens. FIS levels in these PDIOs underestimated functional CFTR rescue, as has been observed before for organoids with SLA levels of over ~40%[15]. Pre-swelling could be explained by the high efficacy of CFTR modulator VX-445 and its dual corrector and potentiator activity[32,33]. Previous drug-induced increase in lumen area was only observed upon combined 24 h corrector/potentiator incubation, likely by rendering the potentiated CFTR protein responsive to endogenous cAMP-PKA signalling[15].

DIS provides an assay format that is able to detect functional CFTR restoration upon VX-445/VX-661/VX-770 treatment. We show significant DIS response to VX-445/VX-661/VX-770 treatment in PDIOs homozygous or compound heterozygous for F508del. However, DIS response to VX-661/ VX-770 treatment in F508del/F508del PDIOs was not significant, while

being effective in vivo[30]. F508del/F508del CFTR restoration by VX-661/VX-770 can be measured by FIS, showing that the FIS assay is a more sensitive assay format and more suitable to measure low to moderate CFTR function rescue, albeit that addition of forskolin during overnight incubation may overcome such lower sensitivities of DIS. Future work should define what levels of forskolin can be used in the DIS assay to most clearly segregate between different functional CFTR levels and associated clinical phenotypes. Importantly, we show that the DIS assay is able to detect significant CFTR rescue in PDIOs where FIS is underestimated due to CFTR modulator induced organoid swelling. These data suggest that the DIS assay platform is a promising new assay for measurement of CFTR modulator efficacy across a diverse set of *CFTR* genotypes.

Taken together, we present an open-source model enabling for the segmentation of PDIOs, in various shapes, from bright-field images, making functional CFTR measurements more accessible and improving standardization (https://github.com/Living-Technologies/OrgaSegment). Additionally, we implement OrgaSegment in the DIS assay, which allows for functional CFTR measurements in PDIOs upon CFTR modulator treatment.

## Methods

### Ethical approval for use of organoids
CF PDIOs with varying *CFTR*-causing mutations were used in this study (exact mutations are specified where used or in supplementary table 1). Organoids used were obtained from the HUB (Hubrecht Organoid Technology) Biobank (www.huborganoids.nl) under TC-Bio protocol number 14-008, or from the UMCU Darmbank under TC-Bio protocol number 19–831, and used according to informed consent.

### Organoid culture
Organoid lines were cultured using standard protocol as described previously[26]. In short, organoids were maintained in 40% matrigel droplets in the presence of WNT-conditioned culturing medium. Medium was refreshed every 2 days and organoids were passaged weekly by manual disruption and reseeding. All organoid cultures were passaged at least 3 times before performing functional CFTR measurements and maintained for a maximum of 20 passages.

### Forskolin induced swelling (FIS) measurements
FIS assays were performed according to standard protocol[26] (supplementary fig. 3a). In short, one day prior to FIS measurements, 7-day-old organoid cultures were harvested and disrupted before seeding them in 96-well culture plates in 4 µL 40% matrigel droplets. Organoids were incubated overnight in 50 µL organoid culture medium, in the presence of correctors VX-661 or VX-445 and VX-661 for VX-661/VX-770 or VX-445/VX-661/VX-770 treated conditions, respectively. After 24 h, calcein green (at a final concentration of 0.84 µM) was added to the organoids before adding potentiator VX-770 (where required) and forskolin. All wells were normalized for DMSO and all CFTR modulator compounds were used at a final concentration of 3 µM. Organoid swelling was monitored using confocal microscopy by imaging every 10 min, over a total time period of 60 minutes, using a Zeiss LSM800 confocal microscope.

### Relative lumen area quantification
Organoids were cultured and seeded as described before. One hour after organoid plating, organoids were imaged using a Leica Thunder widefield microscope (t = 0 hours). Organoids were treated with DMSO or VX-445 and VX-661 and incubated for 24 h. After 24 h, organoids were imaged again and treated with 0.128 µM forskolin and DMSO or VX-770 before imaging one hour later (for a schematic overview of the timeline see Supplementary fig. 3b).

### Drug induced swelling (DIS) measurements
For DIS assays, organoids were cultured as described before. Assay plating was performed following similar protocol as organoid seeding for FIS assays. Organoids were incubated in 100 µL organoid culture medium, for 1–2 h, whereafter organoids were imaged using bright-field microscopy on a Leica Thunder widefield microscope and treated with all CFTR modulator compounds, including potentiators, and forskolin where required, before incubating overnight. The next day, 24 h after organoid treatment, organoids were imaged for the second time, using the same microscope setting as at t = 0 (for a schematic overview of the timeline see supplementary fig. 3c). All DIS measurements are performed in biological triplicates.

### OrgaSegment dataset creation
For the development of the OrgaSegment model we created a dataset containing images of organoids with various degrees of residual CFTR function. Organoids were cultured as described, and after 1 day imaged at different conditions (with or without forskolin and CFTR modulators) with a Zeiss LSM800 confocal microscope using 5x objective and transmitted laser light (ESID) or with a Leica Thunder widefield microscope using 5x objective and bright-field. Images were converted to JPEG, randomized to exclude any experiment information that could influence objective labelling, and uploaded to Labelbox[24] for object labelling. Individual organoids were labelled into a single category (organoid). Labelling was performed by a group of experts, with experience in intestinal organoid culturing. Each image was labelled once and all labels were independently reviewed as quality control, and marked for re-labelling when of insufficient quality. The labelled dataset contained a total of 231 images (15,515 individual organoids), which were randomly split into training, validation and evaluation datasets, yielding datasets of 184 (11,989 organoids), 35 (2552 organoids), and 12 (974 organoids) grayscale images for respectively training, validation, and evaluation.

### OrgaSegment model training
A MASK R-CNN[34] implementation (https://github.com/matterport/Mask_RCNN) was forked and adjusted to optimize parallel GPU usage (https://github.com/Living-Technologies/Mask_RCNN). The ResNet101[35] network was used for feature extraction and the weights trained on the MS COCO dataset[25] was used as a starting point (transferred learning). Training was performed in parallel on 4 Nvidia Quadro RTX6000 GPUs on a SLURM managed HPC-environment. We trained our model on 184 JPEG images for 100 epochs on only the head layers and for an additional 400 epochs on all layers with 50 steps per epoch, batch size of 4 (1 per GPU), and a stochastic gradient descent (SGD) optimizer with a learning rate of 0.001. The images were cropped to 512×512 pixels on-the-fly during training to reduce memory usage. However, the network input image size was not exclusively defined in order to facilitate various image sizes upon model usage. No further data augmentation was performed.

### OrgaSegment model evaluation
In order to evaluate model performance, we predicted the organoid masks of the evaluation dataset and compared the results with the ground-truth (GT), namely the manually labelled organoids of the same dataset. We used the same Average Precision metric and corresponding code as described previously by the Cellpose method[22]. The predictions were matched against the GT at different Intersection over Union (IoU) thresholds ranging between 0.5 and 1.0. A positive match resulted in a true positive (TP), a prediction without any corresponding GT resulted in a false positive (FP), and a GT mask without a prediction resulted in a false negative (FN). For each image the Average Precision (AP) was calculated:

$$AP = \frac{TP}{TP + FP + FN}$$

The AP from each image in the evaluation dataset was averaged, resulting in a single mean Average Precision for every IoU threshold.

## OrgaSegment application

For simple organoid segmentation using the OrgaSegment model we developed an easy-to-use application based on python and the Streamlit app framework. The application requires a Mask-RCNN .h5 model file, a configuration, and JPEG images as input. It will run on both Windows and Linux based systems, either CPU or GPU, with a Anaconda installation and the OrgaSegment anaconda environment.

## Quantification of CFTR modulator response FIS assay

Relative increase in total organoid area (swelling) per well was calculated for the quantification of CFTR functionality, as described before[26]. In short, Zeiss Zen Blue imaging software was used to quantify total organoid area per well, at each time point. Finally, we calculated relative organoid swelling as area under the curve (AUC), normalized to time point 0, as measure of CFTR functionality.

## Quantification of CFTR function using SLA

Total organoid size and luminal size were determined by manual annotation using ImageJ software. SLA was expressed as the percentage of luminal organoid area of the total organoid area:

$$SLA = \left( \frac{luminal\ organoid\ area}{total\ organoid\ area} \right) \cdot 100\%$$

## Quantification of CFTR modulator response DIS assay

Transmitted light microscopy images were exported as JPEG. Images were run on the OrgaSegment application (see OrgaSegment application). Size per organoid per image was calculated and similar organoids at the different timepoints were identified. Swelling per organoid was calculated as follows:

$$Swelling = \frac{Size\ at\ t = 24 hours}{Size\ at\ t = 0 hours}$$

Outliers were identified and excluded, before calculating mean swelling per experimental condition.

## Reporting summary

Further information on research design is available in the Nature Portfolio Reporting Summary linked to this article.

## Data availability

The segmented microscopy dataset that was used to develop OrgaSegment is available at Zenodo (Version 20211206): https://doi.org/10.5281/zenodo.10278229[36]. The analysed data that supports the findings of this study is available at Zenodo: https://doi.org/10.5281/zenodo.10610437[37].

## Code availability

The forked and optimized MASK R-CNN code is available at GitHub (release v2.1.4): https://github.com/Living-Technologies/Mask_RCNN/tree/v2.1.4 (https://doi.org/10.5281/zenodo.7886026)[38]. The OrgaSegment code including configuration, scripts (e.g. training, evaluation, prediction), model weights (.h5 file), application, and README is available at GitHub (release v1.0.1): https://github.com/Living-Technologies/OrgaSegment/tree/v1.0.1 (https://doi.org/10.5281/zenodo.10299615)[39].

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

## Acknowledgements
We thank Labelbox for providing the access for the academic usage of dataset labelling. We thank the members of the Jeffrey Beekman laboratory for labelling organoids. This work was financially supported by the strategic Alliance TU/e, WUR, UU, UMC Utrecht, by the Dutch Cystic Fibrosis Foundation (NCFS) as part of the HIT CF 3.0 program, by the Dutch Rare Diseases Fund, and by the by the Dutch Research Council (NWO) as part of the project Innovative Stem Cell Technology Infrastructure for human organ and disease models (with project number 184.036.006) of the research programme Large-Scale Research Infrastructure.

## Author contributions
J.W.L. designed the study, coordinated the project, performed organoid cultures, imaging experiments, analysed the results, and wrote the manuscript. S.K. and P.J.N. performed organoid cultures, and imaging experiments. M.B.S. analysed results and provided image analysis expertise, N.D.A.N. performed organoid cultures, H.N.S.v.K. maintained the organoid biobank, C.K.E. supervised the research, J.M.B. supervised the research, designed experiments and wrote the manuscript. S.F.B.v.B. provided AI and microscopy expertise, designed and developed the deep-learning segmentation method, wrote the program codes, supervised the research, and wrote the manuscript.

## Competing interests
J.M.B. reports has a patent related to the FIS-assay with royalties paid, and reports personal fees for participation at educational meeting (Chiesi, Jansen), please find full disclosures at https://www.umcutrecht.nl/en/research/researchers/beekman-jm. C.K.v.d.E. report grants from Eloxx, Galapagos NV, Gilead, GSK, Nutricia, ProQR, Proteostasis, Teva, and Vertex Pharmaceuticals Incorporated, and a patent (10006904) with royalties paid. All other authors declare no competing interests. All other authors have nothing to disclose.

## Ethical approval
We declare that the biobank donors gave their informed consent for inclusion before donation. All experiments performed in this manuscript are performed according the this informed consent. The study was conducted in accordance with the Declaration of Helsinki, and the biobank inclusion protocol was approved by the Biobank Ethics Committee of the University Medical Centre Utrecht under protocol numbers 14-008 and 19-831.
