## [Peer Review File · Communications Biology]

Reviewers' comments:

Reviewer #1 (Remarks to the Author):

Lefferts et al., in the current manuscript, report a novel deep-learning-based algorithm that can be utilized for the individual segmentation of intestinal organoids from bright-field images and the quantification of standard assays, such as FIS or DIS. An apparent advantage of this novel algorithm over the existing ones is that it can also handle organoids with different shapes (not preferring round organoids), which is frequently seen among rectal organoids derived from CF patients. My comments are as follows:

Major comments:

1. OrgaSegment can be used to segment individual organoids. Is it possible to set up different clusters based on the shape of the organoids? Can the organoids' shape (or cluster) be correlated with the biological response? There is a heterogeneity in the biological response among organoids, which may also depend on the shape. Extending the manuscript's scope and addressing this issue would be interesting for the field.
2. Based on the data presented in Figure 2. the organoids already show a marked increase in lumen area in response to VX445/VX661, which can not further increase by VX770+forskolin. What could be the reason for that? The combination of the correctors presumably increases the protein level of CFTR in the membrane. However, no activation is happening. Is there an explanation for this prestimulated state? Does the administration of VX770 or forskolin have an effect on these organoids et al.?

Minor comments:

1. What was the maximal passage number of the organoids used during the experiments? The authors mentioned in the methods that the organoids were passaged at least three times before the experiments, but the maximal passage number needs to be mentioned. How long can these organoids be maintained (in passage number)?
2. How long were the organoids cultured between the passage and the initiation of the treatment (Figure 2.)?
3. The dot is missing at the end of the last sentence on page 9.

Reviewer #2 (Remarks to the Author):

Organoid quantification is difficult to complete manually, time-consuming, and subject to bias. Improvements in automation will significantly help expand the utility of this cell culture technology. This work demonstrates a proof of principle automated deep-learning method to help overcome the limitations inherent in the assay.

This work builds on the established literature and improves the automated measurements over other published work as cited in the manuscript. It is of significant relevance to CF although relevance to other disease states is not likely very high.

This work supports the conclusions overall, however, there is a risk of bias inherent in the training used for deep learning, at least as described. See specific critiques below.

Specific critiques.

- 1) Page 4, line 71: The authors state that fluorescent labeling is limited to 3 hours. There are nontoxic fluorescent labels that can be used longer than 3 hours, as well as non-labeled methodology. Please clarify.
- 2) Page 4 line 70: The authors mention segmentation but without detailed definition.
- 3) Were any of the images used in this study previously published elsewhere? If so, please state and cite.
- 4) The authors description of the training of the tool is of significant interest. The authors describe a dataset of >15,000 individually annotated organoids that was manually done by “experts”. What goes in to the original model will heavily bias the tool. The authors must detail what “experts” they mean, how many were employed, did they have 2 or 3 “experts” independently annotating, how did they reconcile differences, and other relevant details. The authors must detail in what way the annotation itself was validated. In the methodology, they state that there were 184 images with 11,989 for training, but far fewer for validation and evaluation. This critique is not necessarily intended to suggest that this is an insufficient number of images, but that the authors should provide more detail and justification as to why this was considered sufficient.
- 5) In methodology, clarify DIS vs. FIS – this terminology is confusing since forskolin and modulator compounds may be required in both according to the provided methodology, and the description of the measurements were very similar excepting frequency of imaging and use of calcein green.
- 6) In results page 6 “CFTR-genotype dependent fluid secretion phenotypes of PDIOs under baseline and CFTR modulator incubation conditions” it is unclear if this was used at any other point for demonstrating OrgaSegment. In this section it is clearly conventional measurements, so why does it need to be here. Recommend clarifying in this section why it belongs here, and also if used for OrgaSegment analysis later.
- 7) In other publications from this author group, various levels of forskolin are used to ensure that responses can be detected appropriately. It looks like only one concentration of forskolin was used, please clarify why.
- 8) There is no limitations section, and this must be rectified.

Rebuttal to reviewers for manuscript COMMSBIO-23-2357A

We would like to express our gratitude to the reviewers for their valuable time and constructive input. Using the reviewer comments we significantly improved the quality of the manuscript. We have addressed all reviewer comments in this document in a point to point reply (clarified in *blue italics*).

Reviewer 1:

Remarks to the Author:

Lefferts et al., in the current manuscript, report a novel deep-learning-based algorithm that can be utilized for the individual segmentation of intestinal organoids from bright-field images and the quantification of standard assays, such as FIS or DIS. An apparent advantage of this novel algorithm over the existing ones is that it can also handle organoids with different shapes (not preferring round organoids), which is frequently seen among rectal organoids derived from CF patients. My comments are as follows:

Major comments:

1. OrgaSegment can be used to segment individual organoids. Is it possible to set up different clusters based on the shape of the organoids? Can the organoids' shape (or cluster) be correlated with the biological response? There is a heterogeneity in the biological response among organoids, which may also depend on the shape. Extending the manuscript's scope and addressing this issue would be interesting for the field.

Thank you for your question, this is indeed an interesting point. We have analysed the shapes of the individual organoid structures in detail using a principal component analysis (PCA), please refer to appendix A (shape analysis) for details and the methods. This analysis yielded 256 different PCA modes. For each mode we analysed the normalized cross correlation between the normalized swelling/growth and the PCA modes. This showed that there was no relation between shape at $t = 0$ and organoid swelling for a specific mode. This suggests that the heterogeneity in biological response is not influenced by the organoid shape at $t = 0$. Additionally, we analysed the changes for each PCA mode compared to the organoid swelling. From these results we can say, generally speaking, organoids that show an increased swelling also become more circular by having a decrease in an elongated PCA mode. At this moment we are not able to draw strong conclusions regarding the relation between organoid shape and swelling response. Since the performed shape analysis does not give us any new biological insights, we have chosen to not include this data into the revised manuscript for publication. It is indeed interesting to investigate in the future, but this might warrant for adjusted image acquisition.

2. Based on the data presented in Figure 2 the organoids already show a marked increase in lumen area in response to VX445/VX661, which cannot further increase by VX770+forskolin. What could be the reason for that? The combination of the correctors presumably increases the protein level of CFTR in the membrane. However, no activation is happening. Is there an explanation for this pre-stimulated state? Does the administration of VX770 or forskolin have an effect on these organoids et al.?

Thank you for your question. The organoids in figure 2 harbour rare CFTR variants, of which the exact mechanism of action is not always known yet. Indeed, the fact that VX445/VX661 therapy already increases relative lumen size, without VX770 and/or forskolin addition, suggests that these variants

have a combined trafficking and potentiation defect which already is restored by the corrective properties of the VX445/VX661 combination, we have added a sentence in the discussion (line 242) to clarify this.

Furthermore, we interpret the mechanisms for the 'pre-swollen' state of organoids in context of pre-treatment as follows. Concerning the activation signal: this likely results from endogenous signalling that is active under standard culture conditions, either through components in the culture media/Matrigel or through auto/paracrine signalling of the intestinal cells. We have never thoroughly investigated the exact nature of this signalling route, but it most likely involve cAMP signalling but potentially also additional signalling conditions (CFTR is a target for cAMP-dependent kinases but also Ca-dependent kinases such as PKC). What is clear is that this phenotype is CFTR dependent: we know that organoids with wtCFTR have fluid filled lumens ('pre-swollen') whereas CF organoids do not (Dekkers et al Science Trans Med, 2016; Cuijx et al Thorax, 2021). Under the pre-swollen state, some additional swelling can be induced by FIS but the relative size increase under such conditions is lower than FIS of CF organoids that are not pre-swollen. This pre-swollen phenotype can also be linked to defined compound pre-treatment and CFTR genotypes. In this context, the potentiators play an important role as potentiators lower the threshold for cAMP to activate CFTR (i.e. they activation properties for potentiated CFTR requires less cAMP (Dekkers et al Science Trans Med, 2016). Under baseline conditions without exogenous forskolin added, compounds that lower the threshold for cAMP activation lead to increase swelling without addition of forskolin, likely by making the restored mutant CFTR resembling the wtCFTR and make it sensitive to the endogenous activation condition. In the current manuscript, we do not study this directly but we can claim that the phenotype is CFTR genotype and CFTR pre-treatment (VX661/VX445) dependent. It is likely conferred by VX445 due to its dual corrector and potentiator mode of action, enabling the VX445/VX661 CFTR mutants to become responsive to endogenous signalling conditions.

Dekkers, J. F. et al. Characterizing responses to CFTR-modulating drugs using rectal organoids derived from subjects with cystic fibrosis. *Sci Transl Med* **8**(344), 344-384 (2016).

Cuyx, S. et al. Rectal organoid morphology analysis (ROMA) as a promising diagnostic tool in cystic fibrosis. *Thorax* **76**, 1146–1149 (2021).

Minor comments:

1. What was the maximal passage number of the organoids used during the experiments? The authors mentioned in the methods that the organoids were passaged at least three times before the experiments, but the maximal passage number needs to be mentioned. How long can these organoids be maintained (in passage number)?

Intestinal organoid cultures can maintain intact stem cell compartments to over 40 passages (Sato et al 2011). Here, the maximum passage we used during these experiments is 20. We added this to the methods section (line 291). We previously showed that CFTR function measurements are not affected by biobanking or in vitro culture for half a year (26 passaging steps, Dekkers et al Science Trans Med, 2016). As such, we do not anticipate any impact of passaging.

Sato, T. et al. Long-term expansion of epithelial organoids from human colon, adenoma, adenocarcinoma, and Barrett's epithelium. *Gastroenterology* **141**, 1762-1772 (2011).

Dekkers, J. F. et al. Characterizing responses to CFTR-modulating drugs using rectal organoids derived from subjects with cystic fibrosis. Sci Transl Med 8(344), 344-384 (2016).

2. How long were the organoids cultured between the passage and the initiation of the treatment (Figure 2.)?

Organoids were treated 1-2 hours after passaging/seeding for experiments. We have added a schematic figure to the methods section to show the timeline of each experiment (see methods section).

3. The dot is missing at the end of the last sentence on page 9.

Thank you, we have adjusted this.

Reviewer 2:

Remarks to the Author:

Organoid quantification is difficult to complete manually, time-consuming, and subject to bias. Improvements in automation will significantly help expand the utility of this cell culture technology. This work demonstrates a proof of principle automated deep-learning method to help overcome the limitations inherent in the assay.

This work builds on the established literature and improves the automated measurements over other published work as cited in the manuscript. It is of significant relevance to CF although relevance to other disease states is not likely very high.

This work supports the conclusions overall, however, there is a risk of bias inherent in the training used for deep learning, at least as described. See specific critiques below.

Specific critiques:

1) Page 4, line 71: The authors state that fluorescent labelling is limited to 3 hours. There are nontoxic fluorescent labels that can be used longer than 3 hours, as well as non-labelled methodology. Please clarify.

Thank you for your comment. We have adjusted the sentence to clarify that fluorescent labelling is limited in time due to leakage of the dye (line 73). Current non-labelled methods have difficulties detecting individual oddly-shaped intestinal organoids (like CF-organoids), creating a need for OrgaSegment, we now point this out in the introduction (line 71)

2) Page 4 line 70: The authors mention segmentation but without detailed definition.

We have adjusted the sentence to clarify segmentation (line 69).

3) Were any of the images used in this study previously published elsewhere? If so, please state and cite.

All images used were created specifically for this study, including the images used for the development of the OrgaSegment model. We have added a sentence to the discussion and methods sections to clarify (line 201 and 329), and as stated in the results section (line 86).

4) The authors description of the training of the tool is of significant interest. The authors describe a dataset of >15,000 individually annotated organoids that was manually done by "experts". What goes in to the original model will heavily bias the tool. The authors must detail what "experts" they mean, how many were employed, did they have 2 or 3 "experts" independently annotating, how did they reconcile differences, and other relevant details. The authors must detail in what way the annotation itself was validated. In the methodology, they state that there were 184 images with 11,989 for

training, but far fewer for validation and evaluation. This critique is not necessarily intended to suggest that this is an insufficient number of images, but that the authors should provide more detail and justification as to why this was considered sufficient.

Thank you for addressing these valid points.

- *We have used a group of organoid experts from our laboratory, including 18 persons for the labelling. All experts have experience with intestinal organoid culturing. All images were labelled by one expert and reviewed by one reviewer, with multiple years' experience with organoid culture. We have added this to the methods section (line 337). As all experts were from the same laboratory, this could potentially induce a bias, we address this now in the limitations section (see point 8).*
- *We used a total of 231 images, containing a total of 15,515 individual organoid structures, for the development of the OrgaSegment model, which were randomly split into training, validation and evaluation datasets, as commonly used following the Pareto principle. 80% of the images (184 images; 11,989 organoid structures) were used for training. The remaining 20% (46 images; 3,526 organoid structures) were used for testing and split into 75% validation (35 images; 2,552 organoid structures) and 25% (12 images; 974 organoid structures) for evaluation. We have adjusted the methods section to clarify this (line 341). The loss values showed smooth monotonic decay during training, and no overfitting was observed by comparing the training and validation loss. Finally, average precision of the model was determined on the evaluation dataset and yielded sufficient values (figure 1C-D).*

5) In methodology, clarify DIS vs. FIS – this terminology is confusing since forskolin and modulator compounds may be required in both according to the provided methodology, and the description of the measurements were very similar excepting frequency of imaging and use of calcein green.

We have added a supplementary figure (Supp figure 3) to the method section with timelines of the different functional assays to clarify the different assay formats.

6) In results page 6 “CFTR-genotype dependent fluid secretion phenotypes of PDIOs under baseline and CFTR modulator incubation conditions” it is unclear if this was used at any other point for demonstrating OrgaSegment. In this section it is clearly conventional measurements, so why does it need to be here. Recommend clarifying in this section why it belongs here, and also if used for OrgaSegment analysis later.

Thank you for this comment. This section was added to illustrate the need for another assay format than the FIS assay in some circumstances where 24hour corrector (VX445/VX661) incubation already leads to swelling of organoids and poses a problem for the FIS assay. We have addressed this in the results section to clarify (line 143 and 152).

7) In other publications from this author group, various levels of forskolin are used to ensure that responses can be detected appropriately. It looks like only one concentration of forskolin was used, please clarify why.

Thank you for this comment. Here, we only used one forskolin concentration to show the concept that DIS is affected also by forskolin, and that addition of forskolin can increase the assay sensitivity to detect functional response. The forskolin concentration used (0.128 μ M) has previously shown to be

relevant for clinical correlations. We have added some additional clarifications in the results and discussion (line 184, 188, and 262).

8) There is no limitations section, and this must be rectified.

This is indeed a valid point. We have now addressed the limitations in the discussion section (line 213).

Appendix A: Intestinal shape analysis for manuscript COMMSBIO-23-2357A

1. Methods:

A principal component analysis (PCA) was done by extracting the boundaries of detected masks. Then a PCA decomposition was performed using the python library Scikit Learn.

1.1

The boundary is a 2D closed polygon where each point is the center of touching edge pixels. The number of points of the boundary is changed 128 points by dividing the total contour length into 128 segments and interpolating the new points from the original curve. The boundary was then oriented with the long axis of the shape aligned along the x-axis. The “long” axis was determined by calculating the 2nd moment matrix of the 2D shape and finding the rotation required to diagonalize the matrix.

The number of points as 128 is based on an observation that the PCA representation didn't show significant improvement beyond 50 points so 128 was chosen because it accurately captures the shapes. The median area of organoids was about 2200 pixels, so 128 is close to two pixels between point, which is more accurate than we need to be (see results).

1.2

The masks were selected from experiments without forskolin from specific donors. They were further filtered by needing to meet the tracking criteria where the organoid was determined to exist in 2 frames. After these criteria were applied there were 20306 masks representing 10153 organoids at two timepoints.

1.3

The PCA calculates the covariance matrix for all of the organoid masks and finds the eigen vectors, which are the principle components. The eigen value is the variance of the specific PCA. With the eigen vectors we can re-write our shape, s , as a sum of the eigen vectors.

$$s = s_0 + \sum_{i=1}^{256} a_i \bar{e}_i$$

Where s_0 is mean shape, a_i is the projection of the shape along each mode, \bar{e}_i .

2. Results & Discussion

The PCA analysis shows some of the results that we expect and helps to exclude some shape features we might have missed.

Figure 1 Shows the first 10 PCA modes, the distribution of projections, and their change in projection vs the growth. By inspection we can see that two modes correlate with the change in volume, the first and the third mode. The first mode reflects the size of an organoid, and corresponds directly with the size. The third mode represents an increase in elongation and correlates negatively with large growth. This represents the swelling that happens. A majority of the organoids grow over time, but the very large growth is predominantly due to swelling. When an organoid swells it becomes rounder and less elongated.

To check if there is a correlation between growth and any of the other PCA modes we plotted the normalized cross correlation between the PCA component and the normalized growth (figure 2).

The normalized growth is defined as:

$$g = (A_1 - A_0)/(A_1 + A_0)$$

Then the cross correlation is calculated as:

$$x_i = \frac{(g - \langle g \rangle)(a_i - \langle a_i \rangle)}{\sigma_g \sigma_{a_i}}$$

with σ_g, σ_{a_i} are the standard deviation of the normalized growth and component projections respectively. The normalized cross correlation only strong correlations with the two modes in the second time point, which we noticed by inspection.

At the level of shape that we are determining, we do not see a correlation in the initial PCA component value to the final result (figure 2). The goal of this study has been to capture area of the organoids after growth for a 24 hour period. There is a wide range of brightness and focal planes for the bright field images which we are able to accurately segment the area. We believe that we cannot make a claim on some of the finer aspects of the shape for the range of conditions we are measuring.

3. Figures

Figure 1:

The first 10 PCA modes and their basic properties. Left is the mean shape plus and minus the corresponding modes. The middle figure is a histogram of the project for each shape along the PCA mode for the two different time points. The right column is a scatter plot of the change in PCA projection value and the growth measured by the ratio of sizes.

Figure 2:

Left shows the normalized cross correlations between the normalized growth and PCA modes. Right is a zoom in to see the different curves for the first 50 modes.

REVIEWERS' COMMENTS:

Reviewer #1 (Remarks to the Author):

The authors addressed my comments properly. I have no further critics.

Reviewer #2 (Remarks to the Author):

The authors have been majorly responsive to all critiques and have provided sufficient detail to better understand their model and for others to utilize this data efficiently to further the field. This manuscript is now acceptable for publication.